# Myelin Disturbances Produced by Sub-Toxic Concentration of Heavy Metals: The Role of Oligodendrocyte Dysfunction

**DOI:** 10.3390/ijms20184554

**Published:** 2019-09-14

**Authors:** Jessica Maiuolo, Roberta Macrì, Irene Bava, Micaela Gliozzi, Vincenzo Musolino, Saverio Nucera, Cristina Carresi, Miriam Scicchitano, Francesca Bosco, Federica Scarano, Ernesto Palma, Santo Gratteri, Vincenzo Mollace

**Affiliations:** 1Institute of Research of Food Safety & Health (IRC-FSH), Department of Health Sciences, University “Magna Graecia”, 88100 Catanzaro, Italy; jessicamaiuolo@virgilio.it (J.M.); robertamacri85@gmail.com (R.M.); irenebava@libero.it (I.B.); micaela.gliozzi@gmail.com (M.G.); xabaras3@hotmail.com (V.M.); saverio.nucera@hotmail.it (S.N.); carresi@unicz.it (C.C.); miriam.scicchitano@hotmail.it (M.S.); francescabosco@libero.it (F.B.); federicascar87@gmail.com (F.S.); palma@unicz.it (E.P.); gratteri@unicz.it (S.G.); 2IRCCS San Raffaele Pisana, 00133 Rome, Italy

**Keywords:** heavy metals, oligodendrocytes, myelination, lipid formation, intracellular calcium regulation

## Abstract

Evidence has been accumulated demonstrating that heavy metals may accumulate in various organs, leading to tissue damage and toxic effects in mammals. In particular, the Central Nervous System (CNS) seems to be particularly vulnerable to cumulative concentrations of heavy metals, though the pathophysiological mechanisms is still to be clarified. In particular, the potential role of oligodendrocyte dysfunction and myelin production after exposure to subtoxic concentration I confirmed. It is ok of heavy metals is to be better assessed. Here we investigated on the effect of sub-toxic concentration of several essential (Cu^2 +^, Cr^3 +^, Ni^2 +^, Co^2+^) and non-essential (Pb^2 +^, Cd^2+^, Al^3+^) heavy metals on human oligodendrocyte MO3.13 and human neuronal SHSY5Y cell lines (grown individually or in co-culture). MO3.13 cells are an immortal human–human hybrid cell line with the phenotypic characteristics of primary oligodendrocytes but following the differentiation assume the morphological and biochemical features of mature oligodendrocytes. For this reason, we decided to use differentiated MO3.13 cell line. In particular, exposure of both cell lines to heavy metals produced a reduced cell viability of co-cultured cell lines compared to cells grown separately. This effect was more pronounced in neurons that were more sensitive to metals than oligodendrocytes when the cells were grown in co-culture. On the other hand, a significant reduction of lipid component in cells occurred after their exposure to heavy metals, an effect accompanied by substantial reduction of the main protein that makes up myelin (MBP) in co-cultured cells. Finally, the effect of heavy metals in oligodendrocytes were associated to imbalanced intracellular calcium ion concentration as measured through the fluorescent Rhod-2 probe, thus confirming that heavy metals, even used at subtoxic concentrations, lead to dysfunctional oligodendrocytes. In conclusion, our data show, for the first time, that sub-toxic concentrations of several heavy metals lead to dysfunctional oligodendrocytes, an effect highlighted when these cells are co-cultured with neurons. The pathophysiological mechanism(s) underlying this effect is to be better clarified. However, imbalanced intracellular calcium ion regulation, altered lipid formation and, finally, imbalanced myelin formation seem to play a major role in early stages of heavy metal-related oligodendrocyte dysfunction.

## 1. Introduction

Heavy metals are present in the Earth’s crust and are released spontaneously and as a consequence of human activities including industrial facilities and anthropic waste procedures mostly connected to the use of fertilisers and pesticides [1]. Generally, heavy metals are very resistant to biological or chemical natural decomposition processes due to their non-degradable nature [2,3]. This phenomenon underlies their accumulation along the food chain through the bio-magnification process [4]. 

Toxicological studies revealed that the heavy metal-related toxicity in mammals depends on several factors, including (1) the rate of absorption, (2) the duration of the exposure, and (3) the cumulative amount absorbed [5,6]. Therefore, heavy metals may accumulate in various vital organs (liver, heart, kidney, brain) and affect their functionalities. Although heavy metal–dependent toxicity is affecting human health leading to a progressive damage of various tissues and functions, neurotoxicity appears to be the common feature of their detrimental effect after their acute and chronic exposure, though the mechanism of action is still to be better clarified. In particular, heavy metals have been shown to interact at cellular level with biological macromolecules such as proteins, lipids, and polynucleotides. In particular, they may produce reactive oxygen species (ROS), undermine the DNA by altering its repair mechanisms, and change the structure and function of some proteins, thus interfering with biological membranes [7]. Moreover, the synthesis of some neurotransmitters may be altered, thus impairing the functioning of the Central Nervous System (CNS) [8,9,10]. On the other hand, heavy metals have been shown to affect myelination process both in developmental brain and in maintaining tissue architecture in adults, thereby leading to neurodegeneration.

Myelination is a physiological process where neuronal axons are coated and isolated with myelin, a multi-lamellar membrane rich in lipids. Myelin speeds up the nerve conduction through the propagation of a saltatory action potential [11]. In all the demyelinating disorders, at the nervous system level, the axon-glia interactions, the neuronal organization, and the electrical conduction are impaired. Chronically demyelinated axons become vulnerable and are destined to a progressive form of disability [12]. The re-myelination process occurs spontaneously; however, in this case, the nerve conduction not always recovers its saltatory nature, and the damages produced by demyelination are not completely repaired [13,14].

In the CNS, oligodendrocytes (OLs) are involved in the synthesis of myelin: more specifically, the membranous part of oligodendrocytes spirally wraps around the segments of the neuronal axon, ensuring rapid transmission of electrical signal in neurons. During the myelination process, oligodendrocytes produce many lipids in a relatively short time [15]. It is possible to assume that, by the end of myelination, oligodendrocytes synthesise about 40% of the total lipids in the brain [16]. Oligodendrocytes (OLs) originate from OL precursor cells (OPCs) [17]; this highly regulated program requires numerous steps to achieve cell differentiation. OPCs proliferate and migrate in other zones of the central nervous system. Finally, the cells differentiated into myelin-forming OLs [18], and this maturation step is accompanied by the expression of specific markers. Proper myelination requires that OLs produce many morphological extensions that will contact with neuronal axons [19]. Myelin is not an inert molecule that simply coats the axon but rather a metabolically active structure: in fact, myelin and the neuronal axon must be regarded as a single metabolically combined functional unit. Oligodendrocyte creates myelin channels—located at the end of the axon—that allow for lipid macromolecules to be transported and moved [20].

Numerous studies in the reference literature show that the exposure to or the intake of high amount of the main heavy metals may lead to the fragmentation and loss of myelin, with demyelinating consequences [21]. Conversely, the effects of heavy metals on the myelinating process, when absorbed in sub-lethal doses, are not known [22].

The present experiments have been designed in order to assess the pathophysiology and the molecular mechanisms underlying the effect of sub-lethal and non-toxic concentrations of several heavy metals in cultured oligodendrocytes alone or in co-culture with neurons. In the last case, this model mimics what happens in one of the segments of the blood–brain barrier (BBB) [23,24].

Since the MO3.13 cells are an immortal human–human hybrid cell line, with the phenotypic characteristics of primary OLs, [25] and resulted from the combination of human rhabdomyosarcoma cells with adult human oligodendrocytes, were differentiated into mature oligodendrocytes, for example, through treatment with the tumor promoter Phorbol 12-myristate 13-acetate (PMA). When the MO3.13 cells are treated with PMA, stopped cell growth and assumed the morphological and biochemical features of mature OLs. In particular we observe the expression of the specific nuclear factor Olig 2 and of the Myelin Basic Protein (MBP) [26]. The use of primary oligodendrocytes is limited by the fact that the cells can no longer be propagated, and for this reason, we decided to use differentiated MO3.13 cell line. Human neuroblastoma SH-SY5Y is a neuronal cell line that has been used for neurotoxicity experiments in vitro [27]. 

In order to maintain oligodendrocyte-neuron interactions, a co-culture system was used and experiments have been performed with both cell lines grown as separately as in co-cultures. For co-culture experiments, specific 12-well Transwell insert plates were used to prevent the cell migration, thanks to a polyester membrane with 1µm pores. In this case, only the growth medium was in contact with both the cell lines as shown in Figure 1. When oligodendrocytes and neurons were grown separately, the obtained results suggested the behavior of the single cell line. On the contrary, when cell lines have been growth in co-culture, results described a condition of cross-talk between oligodendrocytes and neurons.

## 2. Results

### 2.1. Differentiation of the MO3.13 Cell Line 

In order to induce the expression of the phenotypic and metabolic characteristics of mature oligodendrocytes, the cellular differentiation was performed through a treatment with the tumor prometer Phorbol 12-myristate 13-acetate (PMA; Sigma Aldrich, 20151 Milano, Italy) 100 nM for five days. This phorbol ester is involved in cell growth, differentiation and works as a tumor promoter [28,29]. 

At the end of the time established, some phenotypic characteristics of mature oligodendrocytes were assessed. More specifically, the morphological structure of differentiated and non-differentiated oligodendrocytes was compared through confocal microscopy. As shown in Figure 2a, the images of non-differentiated oligodendrocytes (on the left) and differentiated oligodendrocytes (on the right) were compared. The differentiated oligodendrocytes seemed to be adequately stretched from a morphological point of view and show suitable extensions. On the contrary, undifferentiated oligodendrocytes do not develop these extensions [26]. 

Since the MO3.13 cells treated with PMA assumed the morphological and biochemical features of mature oligodendrocytes and OLs synthesize large quantities of lipids, the lipid content was measured. Figure 2b showed that differentiated oligodendrocytes produced more lipids than non-differentiated cells, as highlighted by flow cytometric analysis in which we can see the shift to the right of the peak resulting from the colouring with the specific for lipids Nile Red probe. Finally, a further biomarker of oligodendocyte differentiation was provided by measuring MBP protein expression. (Figure 2c). Differentiated oligodendrocytes were found able to express higher MBP concentration compared to non-differentiated cells, thus confirming that the procedures used throughout the study produced a differentiation of cultured oligodendrocytes. 

### 2.2. The Effect of Heavy Metals on the Viability of Oligodendrocytes and Neurons Grown Either Separately or in Co-Culture 

In order to assess the toxicity of heavy metals when incubated with oligodendrocytes and neurons (MO3.13 and SHSY5Y), cells were plated separately or in co-culture and treated with heavy metals at different concentrations for 24 h. The cell viability was measured through MTT assay. When the cell lines were grown separately, as shown in Figure 3, oligodendrocytes were slightly more sensitive to the action of heavy metals than neurons, especially at higher concentrations.

When the MO3.13 and SHSY5Y cells were grown in co-culture and treated under the same conditions, the neurons were more sensitive to metals, as proved by the related dose-response curve which is constantly lower than the one obtained with oligodendrocytes. This result is shown in Figure 4. After evaluating data obtained under both experimental conditions, a further analysis was carried out by means of a concentration of each heavy metal able to produce a mortality rate not higher than 15–20%. This choice was made to avoid any acute toxic damage to the cells and to achieve a small chronic accumulation of metals, similar to the one due to an unconscious daily intake of these elements. The selected concentrations for each heavy metal are reported in Table 1.

### 2.3. The Effect of Heavy Metals on the Lipid Component in Cells Grown Alone or in Co-Culture 

Incubation of differentiated MO3.13 and SHSY5Y cells grown separately, showed significant differences on lipid content after their exposure to heavy metals (Figure 5a,b). Indeed, MO3.13 cells featured a reduction of cellular lipids after the treatment with heavy metals compared to the related control. The lipid quantification—conducted with two different methods—yielded the same results and the lipid reduction, compared to the control, has been found significant. In contrast, (Figure 6a,b), the treatment of the SH-SY5Y cell line with heavy metals did not show any change in the lipid component compared to the control. This result was obtained through both flow cytometry and spectrometry detection.

When oligodendrocytes and neurons were grown in co-culture and treated with heavy metals (Figure 7a,b), both oligodendrocytes and neurons featured a reduction of the lipid component compared to the related control (except for cadmium and chromium). In this case, the neurons, besides the oligodendrocytes, also had a lower amount of lipids. 

Since the treatment with chromium and cadmium heavy metals did not lead to any significant difference of the lipid component compared to the control in both the cell lines, they were excluded and the study continued with the remaining metals (nickel, aluminium, lead, cobalt, and copper).

In order to understand whether or not the lipid reduction was caused by an altered lipid transport from the oligodendrocytes to the neurons, appropriate measurements in the culture medium in contact with both the cell lines were performed. As shown in Figure 7c, the medium in contact with the oligodendrocytes, after the treatment with each metal, showed significant reduction in lipid content compared to the control. Conversely, the medium in contact with the neurons showed a higher lipid concentration. Lysolecithin (50 µM per 24 h), an efficient demyelinating agent, was used as a positive control to optimize the lipid measurement method in the medium.

### 2.4. The Treatment with Heavy Metals Reduces the MBP Expression in Oligodendrocytes and Neurons Grown in Co-Culture 

In order to verify the effect of heavy metals in MO3.13 and SHSY5Y cells in the formation of myelin, the expression of the main protein that makes up myelin (MBP) in oligodendrocytes and neurons grown in co-culture after the treatment with heavy metals was carried out. As evidenced in Figure 8a and Figure 9a, the MO3.13 and SH-SY5Y lines showed a statistically significant reduction of the MBP expression. In this case, the result was particularly relevant in neurons. Similar responses were obtained through flow cytometry Figure 8b and Figure 9b, in which the shift of fluorescence indicated a variation of MBP expression.

### 2.5. Heavy Metal-Dependent Dysregulation of Cytosolic Calcium

Since modulation of intracellular calcium concentrations has been shown to be involved in both cell lipid synthesis and remodelling, we measured calcium ion changes in MO3.13 and SHSY5Y cells incubated with heavy metals either grown separately or co-cultured. The fluorescent Rhod-2 that binds to the calcium ion, with high affinity, was used as a probe.

Figure 10a shows that treatment with heavy metals resulted in higher concentration of basal cytosolic calcium in MO3.13 cells compared to the control. The treatment with thapsigargin led to an increase in calcium due to the outflow of the ion from the endoplasmic reticulum. Under our experimental conditions, heavy metals had no effects on mitochondrial calcium due to the previous cell treatment with the decoupling of mitochondrial oxidative phosphorylation, carbonyl cyanide-4-(trifluoro-methoxy) phenylhydrazone (FCCP, 1 µM) as well as with the inhibitor of the mitochondrial ATPase, oligomycin, (1 µg/mL). In our experiments, treatment with EGTA culminated with the reduction of calcium concentrations and thapsigargin (100 nM for 8 h) was used to optimize the calcium measurement method.

Figure 10b also shows that in SHSY5Y cells no changes in the concentration of calcium were found after the treatment with heavy metals compared to the control.

When MO3.13 and SHSY5Y cells were grown in co-culture the treatment with heavy metals led to prominent increase of intracellular calcium ion in both the lines, as indicated in Figure 11a,b, respectively. 

## 3. Discussion

The present data investigated, for the first time, on the effect of sub-toxic concentrations of several heavy metals in differentiated oligodendrocytes (MO3.13) and neurons (SHSY5Y), either grown separately or in co-culture. In particular, we assessed that low subtoxic doses of essential (Cu^2+^, Cr^3+^, Ni^2+^, Co^2+^) and non-essential (Pb^2+^, Cd^2+^, Al^3+^) heavy metals, incubated with cells for 24 h, produced changes in cell viability, which was consistent in co-cultured cells. This effect was accompanied by altered lipid and myelin formation, being both effects associated to imbalanced cytosolic calcium ion regulation.

The vulnerability of oligodendrocytes and neurons, when exposed to heavy metals, has been recently assessed thereby re-evaluating the threshold of neuro-toxicity for these products as previously defined [30]. Our data show, in fact, that the toxic responses for all the metals used throughout the study are highlighted when co-cultured oligodendrocytes and neurons are exposed to heavy metals. In addition, our data show that oligodendrocytes are more prone to become dysfunctional after their exposure to heavy metals.

Evidence exists that a bi-directional flow of biochemical signals exists among oligodendrocytes and neurons leading to continuous cell-cell interaction. This modulates several crucial functions including protein and lipid formation in growing cells and ion balance, mostly related to calcium cytosolic concentrations [31]. In addition, myelinated neuronal axons are involved in the normal gene expression of many myelin proteins that are highly affected by heavy metals [32]. Finally, oligodendrocyte-neuron interaction also regulates the oligodendrocyte proliferation and differentiation [33] and many neurological disorders are connected with oligodendrocyte dysfunction [34,35]. In this context, our data confirm that low concentrations of heavy metals, which are unable to produce consistent damage to oligodendrocytes and neurons separately, increase their potential toxic effect when cells are in tight functional connection. In particular, heavy metals tend to produce dysfunctional responses in oligodendrocytes which, at the late stage, lead to damage of both oligodendrocytes and neurons. This confirms previous evidence in which it has been assessed that damaged oligodendrocytes may influence the neurons when cells are able to get functionally connected [33]. This is also confirmed by evidence that oligodendrocyte precursor cells (OPCs) are called out to migrate toward the impairment site and support the axon of the damaged neuron. This process, however, is limited by the number of the endogenous OPCs [36]. If the oligodendrocytes are not regenerated and the myelin is not replaced, the axonal dysfunction may lead to a series of events culminating with neurodegeneration [37] as the one found in several models of heavy metal-induced axonal damage. Furthermore, in some demyelinating neuronal disorders, the re-myelination process is undermined because the OPCs stop differentiating into mature oligodendrocytes, thus leading to an inflammatory state in the nervous system [38]. When included into an in vitro co-culture systems, where OPCs cannot be called out, that damage caused by heavy metals to oligodendrocytes is likely to involve irreversibly neurons. 

This is confirmed by our data. In fact, when the two cell lines were grown separately and treated with increasing concentrations of heavy metals, cells underwent substantial injury which however was observed mainly in oligodendrocytes and only marginally in neurons. Conversely, the growth of the two cell lines in co-culture and the same treatment with heavy metals showed a different behaviour, namely neurons being more vulnerable than oligodendrocytes. These data suggested that the more consistent neuronal susceptibility may be deriving, at least in part, by oligodendrocyte dysfunction, mostly consequent to impaired metabolic support to neurons [39].

On the other hand, the occurrence of an early impairment of oligodendrocytes after exposure to low doses of heavy metals is expressed by the altered cellular lipid component, and MBP biosynthesis found in co-cultured cells. It is known that during the myelin biogenesis, the proteins and lipids are transported to the myelin via a specific vesicular transport system [40]. Moreover, the transport of the vesicles is coordinated with the synthesis of lipids and proteins in order to make myelin complex according to the need of axonal development [41]. Thus, an impaired vesicular transport is supposed to play a role in myelin disturbances that occur when oligodendrocytes are dysfunctional. This seems to occur in our experiments. In fact some lipids destined to the formation of myelin fail to reach their destination, as these molecules were partially found in the culture medium, and this suggests that the lipids starting from oligodendrocytes could not reach the neuronal axons. Thus, when the cells are grown in co-culture, the neurons are directly affected by heavy metal–induced impairment occurring at the early stages in oligodendrocytes.

Finally, low doses of heavy metals used throughout the study have been found to affect calcium ion cytosolic concentration, thereby affecting some relevant functions of oligodendrocytes in their connections with neurons. In particular, evidence exists that several phospholipids are distributed asymmetrically between the two hemilayers of the plasma membrane, and this specific distribution is affecting calcium concentrations in cells, including glial cells and neurons [42]. In our hands, when the cells were grown in co-culture and treated with low amounts of heavy metals, an increase in the calcium ion amount occurred alongside with its simultaneous reduction in the endoplasmic reticulum. These data suggest that low doses of heavy metals lead to early alteration of the cellular organelles in oligodendrocytes and this could explain the simultaneous alteration of MBP and lipid biosynthesis and the subsequent imbalance in calcium ion regulation [43]. These data are in accordance with studies carried out both in vitro and in vivo, in which it has been demonstrated that some heavy metals are responsible for the higher concentration of intracellular calcium with a mechanism independent on the external environment [44,45].

## 4. Materials and Methods

### 4.1. Chemicals

Essential (Cu^2+^, Cr^3+^, Ni^2+^, Co^2+^) and non-essential (Pb^2+^, Cd^2+^, Al^3+^) heavy metals were obtained from Sigma–Aldrich (20151 Milan, Italy). Compounds were solubilized as described [46] and were used to make serial dilutions.

### 4.2. Cell Cultures

The undifferentiated human oligodendrocyte cell line (MO3.13) and the human neuron line (SH-SY5y) have been purchased from the American Type Culture Collection (20099 Sesto San Giovanni, Milan, Italy). The cells were cultured in Dulbecco’s modified Eagle’s medium (DMEM) reinforced with 10% foetal bovine serum (FBS), 100 U/mL penicillin, 100 μg/mL streptomycin, into a humidified 5% CO_2_ atmosphere at 37 °C. Medium was changed every 2–3 days, and when the cell lines reached a 70% confluence, they were treated with essential (Cu^2+^, Cr^3+^, Ni^2+^, Co^2+^) and non-essential heavy metals (Pb^2+^, Cd^2+^, Al^3+^) for 24 h. At the end of the treatment, the cell viability, lipid and myelin contents and calcium ion detection were determined.

To evaluate heavy metal-related damage in both oligodendrocytes and neurons, experiments have been performed with cell lines grown separately or in co-cultures. To this end, specific 12-well Transwell insert plates were used to prevent the cell migration, thanks to a polyester membrane with 1µm pores. Under these experimental conditions, only the growth medium was in contact with both the cell lines as shown in Figure Experimental model Description. Differentiated oligodendrocytes and neurons were placed on plates and left to adhere, and on day 2, they were treated with heavy metals for 24 h at the selected concentrations, as indicated in Table 1.

### 4.3. Cell Differentiation

Since the MO3.13 human cell line used is hybrid and resulted from the combination of human rhabdomyosarcoma cells with adult human oligodendrocytes, cells were distinguished into mature oligodendrocytes.

To this end, the M03.13 oligodendrocytes were treated with Phorbol 12-myristate 13-acetate (PMA; Sigma Aldrich, 20151 Milan, Italy) 100 nM for five days, and at the end of the prescribed time, some phenotypic characteristics were investigated in order to characterize mature oligodendrocytes. 

### 4.4. Lipid Extraction: Flow Cytometric and Spectrometric Quantification

Nile Red is a colouring agent that tightly binds to the cell lipids and, consequently, it was used to quantify these molecules in the cell lines. This colouring agent is not fluorescent when it is dissolved in water or in polar solvents, while it becomes fluorescent when it is dissolved in non-polar solvents (excitation/ emission maxima ~552/636.) At the end of the treatment with heavy metals, oligodendrocytes and neurons (first grown separately and then in co-culture) were picked up and properly incubated with 1 µg / ml of Nile Red dye for 15 min. Later on, the cells were washed in PBS (pH = 7.4) and immediately read to the FACS Accury flow cytometer (Becton Dickinson, 20161 Milan, Italy).

In case of spectrometric assessment of lipids, the cells—after having been treated with heavy metals and incubated with 1 µg/ml of Nile Red dye for 15 min—were picked up and the lipid extraction was performed by means of the Bligh & Dyer method [47]. Then the lipid content related to each sample was subjected to spectrometric reading, and achieved results were interpolated with a straight line built with increasing concentrations and Nile Red notes. The values thus obtained were then normalized for the content of cellular DNA (DNA Quantification kit, Bio Rad, Milan, Italy.) The spectrometric reading was carried out through a VICTOR 2 spectrophotometer (Perkin Elmer, Milan, Italy) at the excitation of 552 nm and the emission of 636 nm.

### 4.5. Phenotypic and Metabolic Characteristics of Mature Oligodendrocytes Measurement by Immunofluorescence

The MO3.13 cells, either differentiated or not, were placed on a plate with a number equal to 10^4^ and, after the adhesion, they were treated with heavy metals. At the end of the treatment, the cells were fixed with 4% paraformaldehyde for 15′ and then permeabilised with 0.1% Triton X-100 for 10′. Later on, they were exposed to the primary antibody (β-Actin; Cell Signalling; Dilution 1:200) for two hours at room temperature in order to assess the cell morphology. The secondary antibody (Goat anti-mouse IgG Dilution 1:400) combined with Alexa Fluor 488 was eventually incubated for one hour. Immunofluorescence analysis was carried out through a confocal microscope (Leica, Italy.)

### 4.6. Cell Growth Assays

MTT assay is based on the detection of live cells with active mitochondria able to reduce 3-(4,5-dimethylthiazol-2-yl)-2,5-diphenyltetrazolium bromide (MTT) to a visible dark-blue formazan, thus providing an indirect measurement of cell viability. After dissemination in 96 well microplates (6 × 10^3^ density), the human MO3.13 oligodendrocytes and human neurons (SH-SY5Y) were grown either separately or in co-culture for 24 h and, subsequently, were treated with heavy metals at different times and concentrations (see Table 1). At the end of the treatment, cells were deprived of medium and incubated with phenol red-free medium containing MTT solution (0.5 mg/mL) for 4h. Finally, 100 μL 10% SDS were added to each well to solubilize the formazan crystals. The plates were gently shaken and the optical density was measured at 570 nm using a spectrophotometer reader. The resulting data were used to calculate cell viability.

### 4.7. Intracellular Calcium Measurements

The Rhodamine 2 indicator (Rhod 2, Molecular Probes) was used to measure intracellular calcium, due to the occurrence that it emits an extensive fluorescence (λ = 581 nm) when it binds to the ion and after having been subjected to excitation λ = 552 nm. The two cell lines (grown either separately or in co-culture) were treated with heavy metals as previously displayed. At the end of the treatment, the cells were exposed to Rhod 2 for 1 h at 25 °C and protected from light. Then, they were washed in a solution free of calcium and magnesium to prevent calcium entering from outside and immediately subjected to flow cytometric analysis by means of a solution free of calcium and magnesium. After the first reading—which provided the basal concentration of the calcium ion—the cells were exposed to thapsigargin (1 µM for 200 s), which increases the calcium cytosolic concentration by releasing the ion from the endoplasmic reticulum. Under these conditions, three new flow cytometric readings were carried out and, after 100 s, ethylene glycol-bis (β-aminoethyl ether)-N,N,N′,N′-tetraacetic acid (EGTA) 500 µM was added to chelate calcium. It is worth remembering that, before the readings, the cells were also treated with the decoupling of the mitochondrial oxidative phosphorylation, carbonyl cyanide-4-(trifluoromethoxy)phenylhydrazone (FCCP, 1 µM), as well as with the inhibitor of the mitochondrial ATPase, oligomycin, (1 µg/mL) Both the substances excluded the involvement of mitochondrial calcium. All flow cytometric detections were carried out by means of a FACS Accury apparatus (Becton Dickinson).

### 4.8. Measurement of the Expression of the Myelin Basic Protein (MBP) through Flow Cytometric Investigation 

The MBP expression was also assessed through flow cytometric investigation. In this case, the two cell lines, grown in co-culture, were treated with heavy metals as described and later picked up and centrifuged. The cells had to be fixed and permeabilised before being treated with the specific antibody, therefore the Cytofix/Cytoperm (BD) solution was used which, simultaneously, fixes and permeabilises the cells. The cells were then suspended once again in 200 µL Cytofix/Cytoperm solution and incubated for 20 min at 4 °C. At the end of the exposure time, the cells were washed with a buffer containing saponin (Perm/Wash buffer, BD) and treated for 2 h with rabbit polyclonal antibody at a 1:50 dilution, for Basic Myelin Protein (MBP, Cell Signalling.) An anti-rabbit conjugated with 1:100 diluted FITC fluorochrome incubated at 4 °C in the dark for 30′ was used as secondary antibody. After appropriate washings, the cells were immediately analysed by means of t flow cytometer detection (FACS Accury, Becton Dickinson).

### 4.9. Statistical Analysis

Data were expressed as mean ± standard deviation (SD) and statistically evaluated for differences using one-way analysis of variance (ANOVA), followed by Tukey-Kramer multiple comparison test (GraphPad software for science).

## 5. Conclusions

In conclusion, our data show that low doses of several heavy metals is accompanied by early dysfunction in oligodendrocytes, which appear to be more vulnerable than neurons when grown separately. Co-culturing both cell types is accompanied by consistent increase of heavy metal–related damage, as reflected by reduced viability of both cell types, reduced lipid and MBP formation, and calcium ion dysregulation, thus confirming the contribution of dysfunctional oligodendrocytes in making neurons more vulnerable to subtoxic concentration of heavy metals. Taken together, our data suggest that heavy metals, even used at low concentration, may interfere with the cross-talk between oligodendrocytes and neurons, an effect that could play a role in some neurodegenerative disorders. 

## Figures and Tables

**Figure 1 ijms-20-04554-f001:**
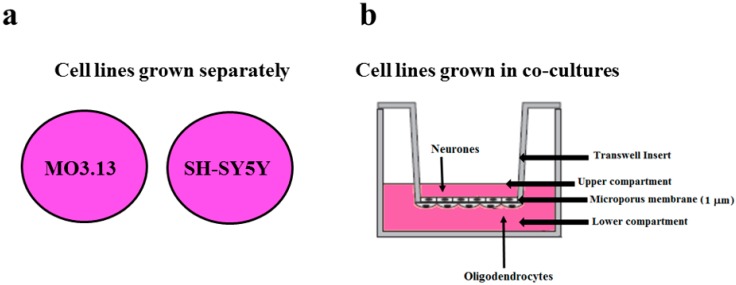
(**a**) Cell lines were grown separately (**b**) Differentiated oligodendrocytes and neurons were placed on the outer and the inner portion of Transwell insert respectively. The used Transwell insert may prevent the cell migration thanks to a polyester membrane with 1µm pores. Under these experimental conditions, only the growth medium was in contact with both the cell lines.

**Figure 2 ijms-20-04554-f002:**
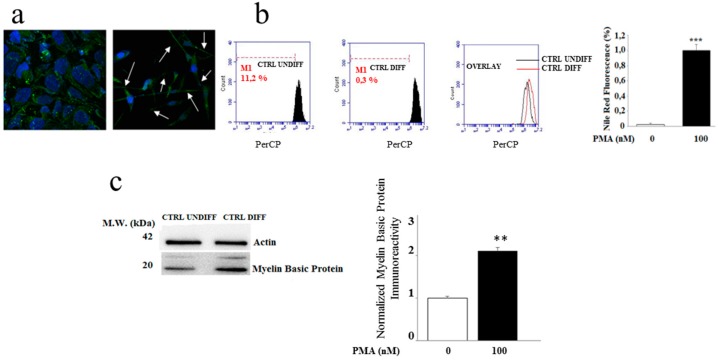
Differentiation of the MO3.13 cell line (**a**) Morphological structure of oligodendrocytes through confocal microscopy. The MO3.13 cell line was differentiated by means of treatment with Phorbol 12-myristate 13-acetate 100 nM for five days. At the end of the treatment, the cells were exposed to the β-actin primary antibody in order to assess the cell morphology. On the left of panel (**a**), there are undifferentiated oligodendrocytes; on the right of panel (**a**) there are differentiated oligodendrocytes. The white arrows indicate the extensions formed following the differentiation of MO3.13. (**b**) The lipid quantification, both in differentiated and in non-differentiated oligodendrocytes, through flow cytometric analysis. The oligodendrocytes were incubed with Nile Red. The overlapping of fluorescence peaks, obtained through the analysis of undifferentiated and differentiated cells, is shown in the overlay panel. On the right of panel (**b**), the quantification of the flow cytometric reading is shown. (**c**) Expression of the Myelin Basic Protein (MBP) through western blotting. On the right of panel (**c**), the quantification of the bands thus obtained is indicated. Values of three independent experiments are expressed as mean ± standard deviation (sd). ** denotes *p* < 0.01 vs. control; *** denotes *p* < 0.001 vs. control; Analysis of Variance (ANOVA) followed by the Tukey-Kramer comparisons test).

**Figure 3 ijms-20-04554-f003:**
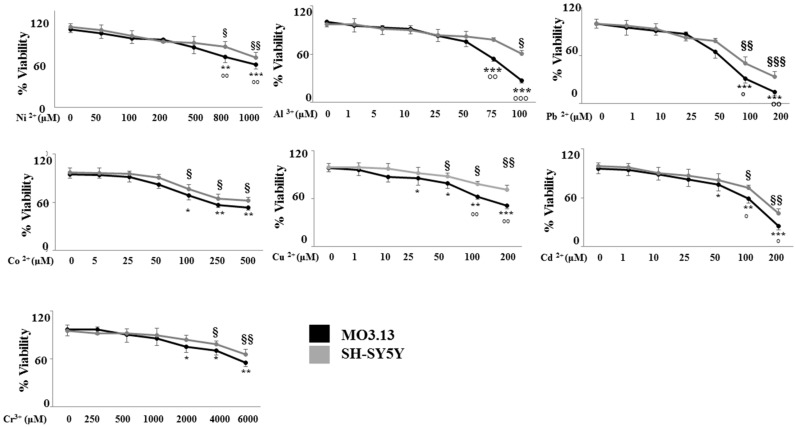
Analysis of cell viability of oligodendrocytes and neurons, grown separately, after the treatment with different concentrations of heavy metals. Human differentiated oligodendrocytes (MO3.13) and human neurons (SH-SY5Y) were grown separately and treated with heavy metals for 24 h at the concentrations reported in the figure. The cell viability was measured through 3-(4,5-dimethylthiazol-2-yl)-2,5-diphenyltetrazolium bromide (MTT) assay. Values of three independent experiments are expressed as mean ± standard deviation (sd). * denotes *p* < 0.05 vs. control; ** denotes *p* < 0.01 vs. control; *** denotes *p* < 0.001 vs. control. ^§^ denotes *p* < 0.05 vs. control; ^§§^ denotes *p* < 0.01 vs. control; ^§§§^ denotes *p* < 0.001 vs. control; ° denotes *p* < 0.05 vs. oligodendrocyte viability under the same treatment conditions; °° denotes *p* < 0.01 vs. oligodendrocyte viability under the same treatment conditions; °°° denotes *p* < 0.001 vs. oligodendrocyte viability under the same treatment conditions. Analysis of Variance (ANOVA) followed by the Tukey-Kramer comparisons test.

**Figure 4 ijms-20-04554-f004:**
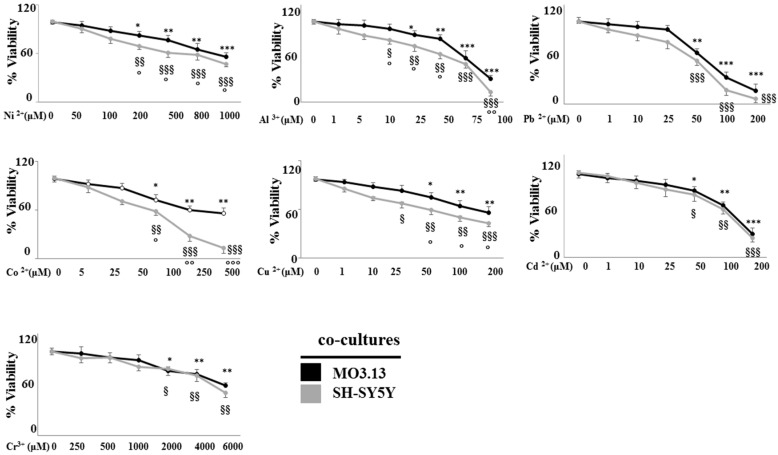
Analysis of cell viability of differentiated oligodendrocytes and neurons grown in co-culture, after the treatment with different concentrations of heavy metals for 24 h. The cell viability was measured through the MTT assay. Values of three independent experiments are expressed as mean ± standard deviation (sd). * denotes *p* < 0.05 vs. control; ** denote *p* < 0.01 vs. control; *** denote *p* < 0.001 vs. control; ^§^ denotes *p* < 0.05 vs. control; ^§§^ denote *p* < 0.01 vs. control; ^§§§^ denote *p* < 0.001 vs. control. ° denotes *p* < 0.05 vs. oligodendrocyte viability under the same treatment conditions; °° denote *p* < 0.01 vs. oligodendrocyte viability under the same treatment conditions; °°° denote *p* < 0.001 vs. oligodendrocyte viability under the same treatment conditions Analysis of Variance (ANOVA) followed by the Tukey-Kramer comparisons test).

**Figure 5 ijms-20-04554-f005:**
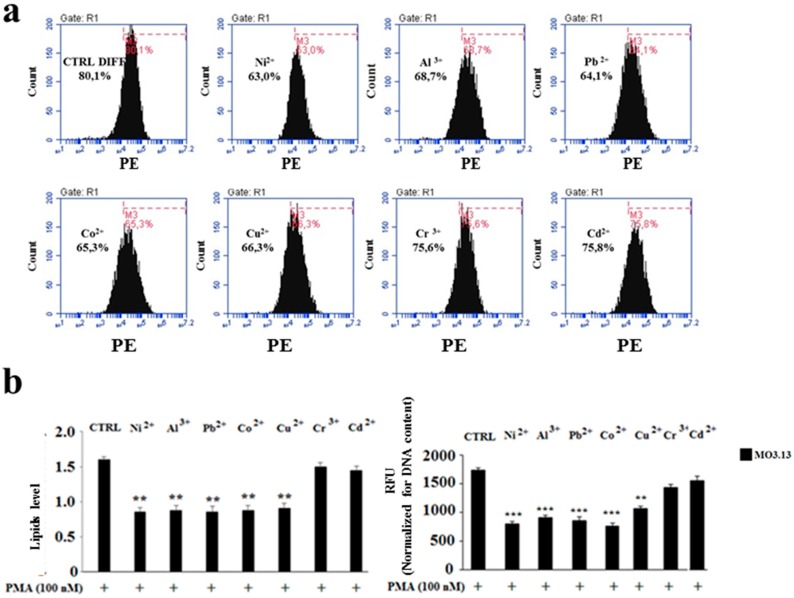
Oligodendrocyte lipid quantification through flow cytometric and spectrophotometric analysis. Differentiated human oligodendrocytes and human neurons were grown separately and treated with heavy metals for 24 h. At the end of the treatment, the cells were picked up and incubated at 37 °C with Nile Red and analysed with the FACS Accury flow cytometer. For the purposes of the spectrometric assessment of lipids, the cells, after the treatment with heavy metals and incubation with Nile Red colouring agent, were picked up and lipids were extracted through the Bligh & Dyer method. Then, the lipid content relating to each sample was subjected to spectrometric reading. The values thus obtained were normalised for the cellular DNA content. Panel (**a**) denotes representative plots relating to the flow cytometric reading of oligodendrocytes with the corresponding marker which shows the shift of the peaks to the right or left. Panel (**b**) on the left indicates the quantification of the flow cytometric analysis of oligodendrocytes; on the right, the values, which were obtained through the spectrometric analysis, are reported. Values of three independent experiments are expressed as mean ± standard deviation (sd). ** denotes *p* < 0.01 vs. differentiated control; *** denotes *p* < 0.001 vs. differentiated control. Analysis of Variance (ANOVA) followed by the Tukey-Kramer comparisons test.

**Figure 6 ijms-20-04554-f006:**
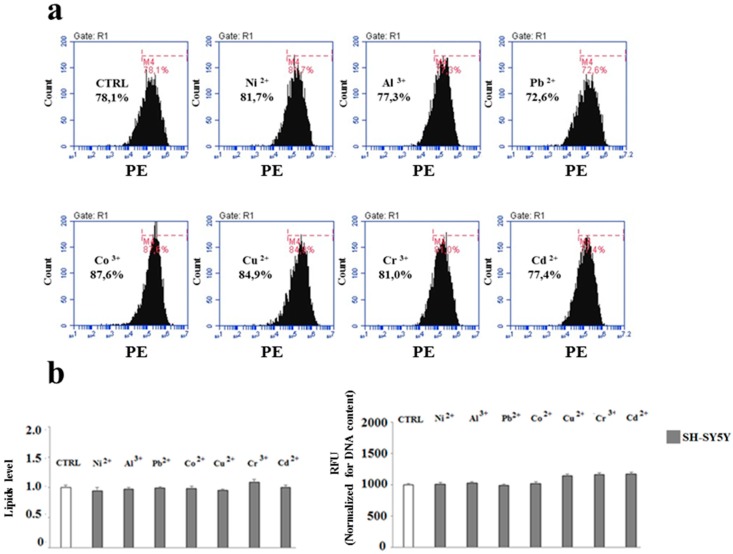
Neuron lipid quantification through flow cytometric and spectrophotometric analysis. Differentiated human oligodendrocytes and human neurons were grown separately and treated with heavy metals for 24 h. At the end of the treatment, the cells were picked up and incubated at 37 °C with Nile Red and analysed with the FACS Accury flow cytometer. For the purposes of the spectrometric assessment of lipids, the cells, after the treatment with heavy metals and incubation with Nile Red colouring agent, were picked up and lipids were extracted through the Bligh & Dyer method. Then, the lipid content relating to each sample was subjected to spectrometric reading. Panel (**a**) denotes the plots relating to the flow cytometric reading of neurons with the corresponding marker which shows the absence of a shift of the peaks to the right or left. Panel (**b**) on the left indicates the quantification of the flow cytometric analysis; on the right, the values obtained through the spectrometric analysis are reported. The values thus obtained were then normalised for the cellular DNA content. Values of three independent experiments are expressed as mean ± standard deviation (sd). No significant differences were detected among different heavy metal treatments. Analysis of Variance (ANOVA) followed by the Tukey-Kramer comparisons test.

**Figure 7 ijms-20-04554-f007:**
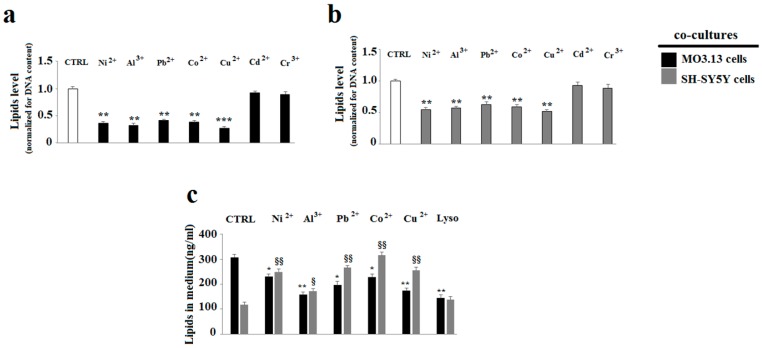
Lipid quantification through spectrometric analysis. Differentiated human oligodendrocytes and human neurons (SH-SY5Y) were grown in co-culture and treated with heavy metals for 24 h. At the end of the treatment, the cells were incubated with Nile Red. Then, lipids were extracted through the Bligh & Dyer method. The lipid content relating to each sample was subjected to spectrometric reading. The values thus obtained were then normalised for the cellular DNA content. Panel (**a**) shows achieved results through the spectrometric analysis carried out on oligodendrocyte lipids. Panel (**b**) indicates the values obtained through spectrometric analysis carried out on neuron lipids. Panel (**c**) reports the achieved results through spectrometric analysis carried out on the lipids found in the media in contact with the two cell lines grown in co-culture. Lysolecithin 50 µM per 24h was used as a positive control for demyelination. Values of three independent experiments are expressed as mean ± standard deviation (sd). * denotes *p* < 0.05 vs. control; ** denotes *p* < 0.01 vs. control; *** denotes *p* < 0.001 vs. control. ^§^ denotes *p* < 0.05 vs. control (grey bars); ^§§^ denotes *p* < 0.01 vs. control (grey bars). Analysis of Variance (ANOVA) followed by the Tukey-Kramer comparisons test.

**Figure 8 ijms-20-04554-f008:**
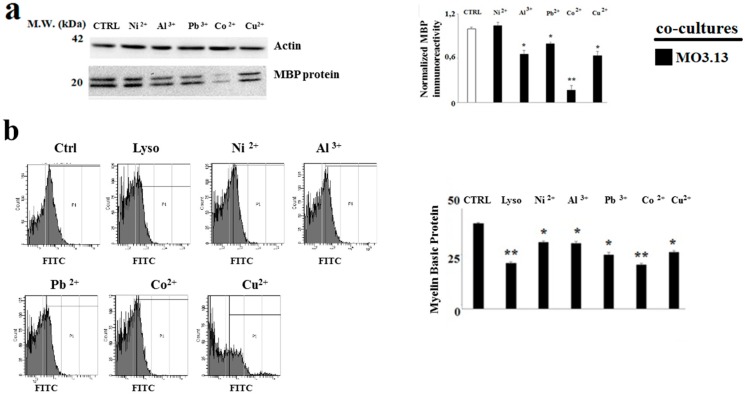
Co-culture-derived oligodendrocytes. Measurement of expression of Myelin Basic Protein (MBP) through Western blotting and flow cytometric analysis. Differentiated human oligodendrocytes and human neurons were grown in co-culture and treated with heavy metals for 24 h. At the end of the treatment, the total lysates of the samples were subjected to western blotting analysis. In case of flow cytometric analysis, at the end of the treatment, the cells were fixed and permeabilized in the Cytofix/Cytoperm solution and treated with rabbit polyclonal antibody for Myelin Basic Protein. As a secondary antibody, it was decided to use an anti-rabbit conjugated with fluorophore FITC. Panel (**a**) shows the MBP expression in oligodendrocytes under the indicated conditions. The quantification is visible on the right of the panel. Panel (**b**) shows the MBP expression in oligodendrocytes obtained through flow cytometric analysis. The quantification is visible on the right of the panel. Values of three independent experiments are expressed as mean ± standard deviation (sd). * denotes *p* < 0.05 vs. control; ** denotes *p* < 0.01 vs. control. Analysis of Variance (ANOVA) followed by the Tukey-Kramer comparisons test.

**Figure 9 ijms-20-04554-f009:**
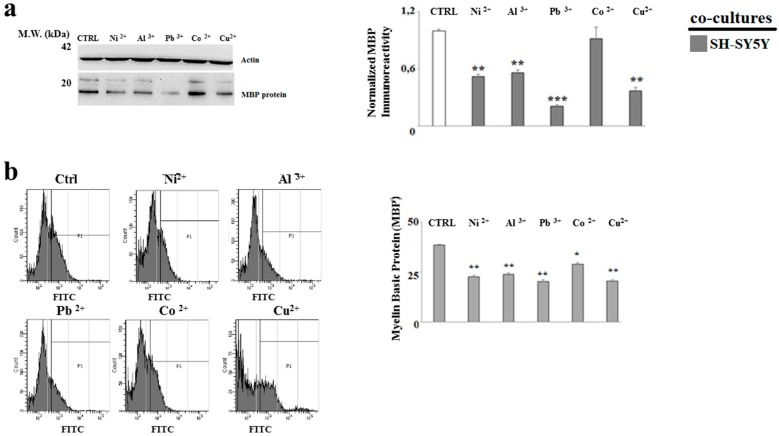
Co-culture-derived neurons. Measurement of expression of Myelin Basic Protein (MBP) through Western blotting and flow cytometric analysis. Differentiated human oligodendrocytes and human neurons were grown in co-culture and treated with heavy metals for 24 h. At the end of the treatment, the total lysates of the samples were subjected to western blotting analysis. In case of flow cytometric analysis, at the end of the treatment, the cells were fixed and permeabilized in the Cytofix/Cytoperm solution and treated with rabbit polyclonal antibody for Myelin Basic Protein. As a secondary antibody, it was decided to use an anti-rabbit conjugated with FITC fluorochrome. Panel (**a**) shows the MBP expression in neurons under the indicated conditions. The quantification is visible on the right of the panel. Panel (**b**) shows the MBP expression in neurons obtained through flow cytometric analysis. The quantification is visible on the right of the panel. Values of three independent experiments are expressed as mean ± standard deviation (sd). * denotes *p* < 0.05 vs. control; ** denotes *p* < 0.01 vs. control, *** denotes *p* < 0.001 vs. control. Analysis of Variance (ANOVA) followed by the Tukey-Kramer comparisons test.

**Figure 10 ijms-20-04554-f010:**
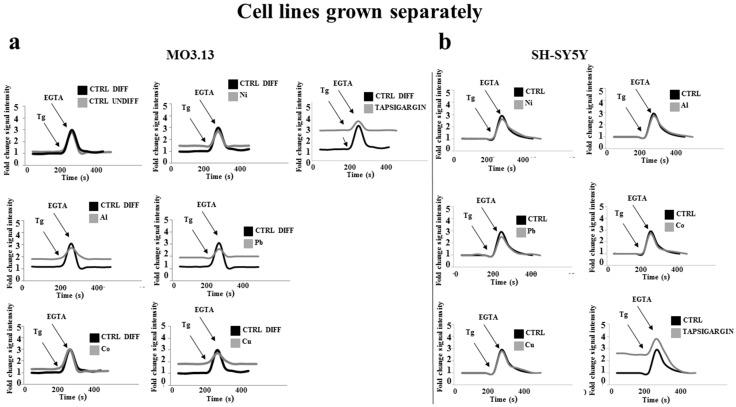
Measurement of cytosolic calcium in oligodendrocytes and neurons grown separately. Differentiated human oligodendrocytes and human neurons were grown separately and treated with heavy metals for 24 h. At the end of the treatment, the cells were exposed to Rhod 2 and subjected to flow cytometric analysis. The cells were then exposed to 1 µM thapsigargin for 200 s and a new flow cytometric reading was performed. After 100 s, ethylene glycol-bis(β-aminoethyl ether)-N,N,N′,N′-tetraacetic acid (EGTA) 500 μM was added to chelate calcium. A new flow cytometric reading was performed. Panel (**a**) shows the curves built on the achieved results from the flow cytometric readings performed on oligodendrocytes under the experimental conditions described above. Each graph refers to a different metal as shown in the figure. Panel (**b**) shows the curves built on the values obtained from the flow cytometric readings performed on neurons under the experimental conditions described above. Each graph refers to a different metal as evidenced in the figure. Values of three independent experiments are indicated. A representative image is shown. Analysis of Variance (ANOVA) followed by the Tukey-Kramer comparisons test.

**Figure 11 ijms-20-04554-f011:**
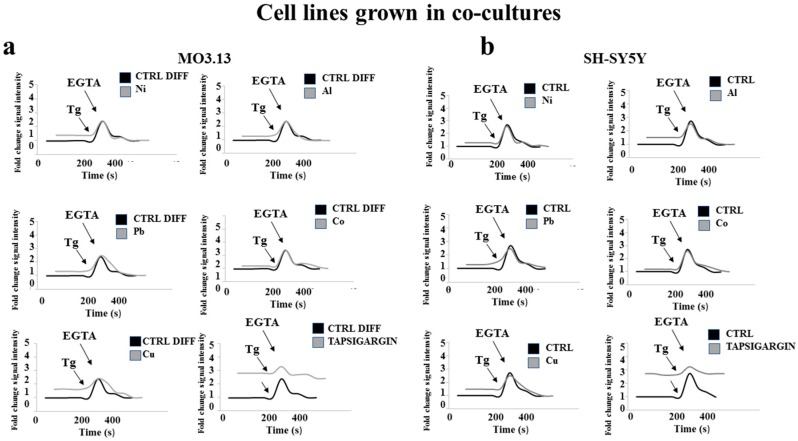
Measurement of the cytosolic calcium of oligodendrocytes and neurons grown in co-culture. Differentiated human oligodendrocytes and human neurons were grown in co-culture and treated with heavy metals for 24 h. At the end of the treatment, the cells were exposed to Rhod 2 and subjected to flow cytometric analysis. The cells were then exposed to 1 µM thapsigargin for 200 s and a new flow cytometric reading was performed. After 100 s, ethylene glycol-bis(β-aminoethyl ether)-N,N,N′,N′-tetraacetic acid (EGTA) 500 μM was added to chelate calcium. A new flow cytometric reading was conducted. Panel (**a**) shows the curves built on the achieved results from the flow cytometric readings performed on oligodendrocytes in the experimental conditions described above. Each graph refers to a different metal as shown in the figure. Panel (**b**) shows the curves built on the values obtained from the flow cytometric readings performed on neurons under the experimental conditions described above. Each graph refers to a different metal as evidenced in the figure. Values of three independent experiments are indicated. A representative image is shown. Analysis of Variance (ANOVA) followed by the Tukey-Kramer comparisons test.

**Table 1 ijms-20-04554-t001:** The selected concentrations for each heavy metal.

METAL	USED DOSE
Nickel (Ni^2+^)	200 µM
Aluminum (Al3^+^)	25 µM
Lead (Pb^2+^)	25 µM
Cobalt (Co^2+^)	25 µM
Copper (Cu^2+^)	10 µM
Cadmium (Cd^2+^)	25 µM
Chrome (Cr ^3+^)	1000 µM

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
