# Peer review of "Myelin Disturbances Produced by Sub-Toxic Concentration of Heavy Metals: The Role of Oligodendrocyte Dysfunction"

_ijms, 2019, doi:10.3390/ijms20184554_

Round 1

Reviewer 1 Report

Reviewer comments:

This study describes the effects of sub-toxic concentrations of various heavy metals on oligodendrocyte and neuronal cell lines, showing that there are significant functional effects on both types of cell lines, with intracellular calcium ion regulation, altered lipid formation and imbalanced myelin formation as key observations leading to the authors conclusions. This study provides valuable information regarding sub-toxic concentrations of heavy metals on numerous neuronal like cell types, however there are a number of minor and major concerns that will need to be addressed prior to publication. Some English editing for grammar in sections of the manuscript are provided below. When referring to MO3.13 cells they should be clearly stated, MO3.13 cell are not oligodendrocytes, they have oligodendrocyte like characteristics. ie. Line 111-113 referring to differentiated MO3.13 cells as oligodendrocytes is misleading…this needs to be corrected and consistent throughout the entire manuscript. i.e. oligodendrocyte-like cells. This manuscript draft needs significant improvement.

Line 51. Extra space in front of “In particular,…”

Line 87. “was performed using….PMA”

Line 94. chunky and indistinct is not accepted cell morphology / terminology …please revise

Line 96. Please clarify between differentiated, non-differentiated and mature cells, or make terminology consistent.

Line 106. Confocal image in fig 1. Is very poor and needs to clearly show morphological differences with use of arrows.

Line 113. The oligodendrocytes were cubed? Maybe “Incubated”?

Line 125: Please explain the rationale for using independent and co-cultured experiments.

Line 139: Growth curves? Do you mean analysis of cell viability?

Line 146. Please explain vitality…do you mean viability?

Line 193. Spectrometric?...should be Spectrophotometric?

Line 223. Spectrometric?...should be Spectrophotometric?

Line 255. FITC is a “fluorophore”

Line 263. Neurones…the rest of the manuscript “neurons” please select one and keep consistent.

Major concerns:

The introduction is lacking adequate description of both cell lines as oligodendrocyte and neuronal cells, for example the myelinating potential of MO3.13 cells, given that myelination capacity is a central focus of some of the experiments. The context of using immortalised cell lines rather than primary cells needs to be clear, given that immortalised cell lines may have their own idiosyncratic behaviours in-vitro. The significance of the co-culture experiments is not explained, neither is there a mention of what conditions of co-culture of these 2 cell lines have on their lipid content over time.  Most of the figures including figs 2, 3 9 and 10 were poorly legible, and need to be made clearer. Many of the figures contain information that is more suitable to be included in the method section, providing more space to adequately explain the data in each figure. More importantly Flow cytometry axes are not legible, FL2-A should state the fluorophore that is being measured. Figure 7A missing MW labels for proteins Too many instances of nomenclature inconsistencies to be itemised. Please carefully revise manuscript. Recommend major changes, clearly stating the purpose of each experiment and clearly relating each finding to their discussion points.

Author Response

First of all thank you for your suggestion. The paper has been revised as requested and changes are shown below.

Line 51. Extra space in front of “In particular,…” It was correct Line 51

Line 87. “was performed using….PMA” In order to induce the expression of the phenotypic and metabolic characteristics of mature oligodendrocytes, the cellular differentiation was performed through a treatment with the tumor prometer Phorbol 12-myristate 13-acetate (PMA; Sigma Aldrich) 100 nM for 5 days. This phorbol ester is involved in cell growth, differentiation and works as a tumor promoter [22, 23].

Line 94. chunky and indistinct is not accepted cell morphology / terminology …please revise. The differentiated oligodendrocytes seemed to be adequately stretched from a morphological point of view and show suitable extensions. On the contrary, undifferentiated oligodendrocytes do not develop these extensions (lines 115-117). In the caption of figure 1 the sentence has been added "The white arrows (panel a on the right) indicate the extensions formed following the differentiation of MO3.13 (lines 134-135).

Line 96. Please clarify between differentiated, non-differentiated and mature cells, or make terminology consistent. A better description has been added in introduction.

Line 106. Confocal image in fig 1. Is very poor and needs to clearly show morphological differences with use of arrows. DONE.

Line 113. The oligodendrocytes were cubed? Maybe “Incubated”?  Done, line 142.

Line 125: Please explain the rationale for using independent and co-cultured experiments. Done, Line 100-102.

Line 139: Growth curves? Do you mean analysis of cell viability? Has been corrected line 170.

Line 146. Please explain vitality…do you mean viability? DONE line 173

Line 193. Spectrometric?...should be Spectrophotometric? DONE, line 219

Line 223. Spectrometric?...should be Spectrophotometric? DONE, line 237

Line 255. FITC is a “fluorophore” Has been CORRECTED LINE 286

Line 263. Neurones…the rest of the manuscript “neurons” please select one and keep consistent. Has been CORRECTED LINE 295

Reviewer 2 Report

Authors investigated on the effect of sub-toxic concentration of several essential (Cu2 +, Cr3 +, Ni2 +, Co2+) and non-essential (Pb2 +, Cd2 +, Al3 +) heavy metals on MO3.13 and SHSY5Y human oligodendrocyte and neuronal cell lines (grown individually or in co-culture).

This research is very interesting and yields important information for researchers and clinicians in this field. Therefore, finding mechanisms would be a useful therapeutic approach for neurodegenerative disorders. The authors are advised the points below.

The sections in the Introduction need to write purpose of this study. Better to have schematic diagram in the last figure to understand easily for Reviewer’s.

Author Response

Thank you for your suggestion. The paper has been revised as requested.

The introduction has been modified including adequate description of both cell lines as oligodendrocyte and neuronal cells, Figure that explains the experimental model has been added. The purpose of this study was added. Graphical Abstract has been included.